# Comparing posttraumatic growth in mothers after stillbirth or early miscarriage

**Kirsty Ryninks**[1], **Megan Wilkinson-Tough**[1], **Sarah Stacey**[2], **Antje Horsch**[3,4]*

**1** Department of Psychology, University of Bath, Bath, United Kingdom, **2** St Michael's Hospital, University Hospitals Bristol NHS Foundation Trust, Bristol, United Kingdom, **3** Institute of Higher Education and Research in Healthcare, University of Lausanne, Lausanne, Switzerland, **4** Department Woman-Mother-Child, Lausanne University Hospital, Lausanne, Switzerland

\* antje.horsch@chuv.ch

**Data Availability Statement:** All data pertaining to this study are uploaded to the University of Bath repository: Ryninks, K., Wilkinson-Tough, M., Stacey, S., Horsch, A. (2022). Dataset for:

## Abstract

The possibility of posttraumatic growth in the aftermath of pregnancy loss has received limited attention to date. This study investigated posttraumatic growth in mothers following stillbirth compared to early miscarriage. It was hypothesised that mothers following stillbirth will demonstrate more posttraumatic growth, challenge to assumptive beliefs, and disclosure than mothers following early miscarriage. The study also sought to understand how theoretically-derived variables of the Model of Growth in Grief (challenge to assumptive beliefs and disclosure) explained unique variance in posttraumatic growth when key factors were controlled for. One-hundred and twenty women who had experienced a stillbirth (N = 57) or early miscarriage (N = 63) within the last two to six years completed validated questionnaires in an online survey relating to posttraumatic growth and key variables relevant to emotional adjustment post-bereavement. Participants who had experienced a stillbirth demonstrated significantly higher levels of posttraumatic growth, posttraumatic stress symptoms, perinatal grief, disclosure, challenge to assumptive beliefs and rumination than participants who had experienced an early miscarriage (Cohen's *d* ranged .38-.94). In a hierarchical stepwise regression analysis, challenge to assumptive beliefs alone predicted 17.5% of the variance in posttraumatic growth. Intrusive and deliberate rumination predicted an additional 5.5% of variance, with urge to talk, reluctance to talk, and actual self-disclosure predicting a further 15.3%. A final model including these variables explained 47.9% of the variance in posttraumatic growth. Interventions targeting challenge to assumptive beliefs, disclosure, and rumination are likely to be clinically useful to promote psychological adjustment in mothers who have experienced stillbirth and early miscarriage.

## Introduction

Losing a child is one of the most devastating things that a parent can experience in their lifetime [1]. Expectant parents develop commitment to their unborn child that alters how they view themselves and the developing baby [2]. The loss therefore entails the loss of anticipated joy and motherhood, as well as the struggle to identify as a mother without the presence of a

"Comparing posttraumatic growth in mothers after stillbirth or early miscarriage". Bath: University of Bath Research Data Archive. https://doi.org/10.15125/BATH-01173.

**Funding:** The authors received no specific funding for this work.

child [3,4]. Mothers may experience a variety of losses in pregnancy, including early miscarriage (EM; during first trimester of pregnancy up to 12 weeks), late miscarriage (LM; during second trimester at 13–23 weeks) and stillbirth (SB; defined as intrauterine death after 24 weeks gestation in the United Kingdom [5]). Miscarriage affects 200,000 couples every year in the UK and 85% of all miscarriages occur in the first trimester [6]. The global prevalence rate of SB in 2015 was 2.7 million [7] and SB is estimated to occur in nearly 1 in 200 pregnancies [8]. In comparison to other types of child loss, pregnancy loss is often treated as less significant [9]. As the loss may not always be recognised socially, parents can be left to grieve in isolation which may affect their psychological adjustment [10].

## Perinatal loss outcomes

Losing a baby through miscarriage [11] or SB [12] is recognised as traumatic [13] and can result in a variety of psychological reactions [14,15]. Perinatal loss has been found to result in post-traumatic stress disorder (PTSD) in 25% of mothers, with levels of symptom severity at one-month post-loss that are equivalent to that found in other traumatised populations [16]. Bereavement reactions have been reported to be pervasive, powerful and complex, as illustrated by research into mothers' experience of stillbirth [17–20]. Mothers are often unprepared for a perinatal loss and experience a range of psychological reactions, including denial, guilt, anger, grief, and feelings of 'empty arms' [17,21–23]. Although the majority of mothers adjust and regain a sense of purpose, 15–25% seek additional support for enduring adjustment difficulties in the year after their loss [3,24], with symptoms of depression, anxiety [25], post-traumatic stress [12,16] and affective disturbance in subsequent pregnancy [26–28] identified for a significant proportion of mothers following SB.

Most studies to date have grouped together different types of perinatal losses in their efforts to understand the psychological impact of loss in pregnancy. However, there are a number of differences between EM and SB and studies have highlighted that the psychological impact of these differing losses requires clarification [29,30]. Where loss in pregnancy is experienced early on, the loss often remains unacknowledged outside of the immediate family [31,32]. In contrast, as SB occurs after 24 weeks gestation, the pregnancy is visible and will have been discussed across the family's social and professional networks. Grouping types of perinatal loss could be unhelpful [18] and result in inconsistency in findings. Early research aiming to understand the impact of loss at different stages of pregnancy found that attachment to the unborn child and grief reactions were greater following loss of further progressed pregnancies [33]; however, Lovell [34] has argued that it is commitment to the pregnancy and meaning of the loss that matters, rather than pregnancy duration.

## Posttraumatic growth

While the death of a loved one is a devastating and painful experience, coping with bereavement may also provide a context for significant positive change [35–37]. For example, following SB, a sample of mothers described an increased sense of empowerment to challenge the medical profession, improve bereavement care, and raise awareness of SB [38]. Posttraumatic Growth (PTG) is defined as "positive psychological change experienced as a result of the struggle with highly challenging life circumstances" [39, p.157]. Through the journey to make sense of the world post-trauma, individuals may develop a greater appreciation for their relationships, gain spiritual insights, recognise their strengths, and develop acceptance that it is not always possible to prevent negative outcomes. Any positive changes arising in response to the trauma occur in conjunction with grief and distressing emotions, not in place of them [40]. The phenomenon of PTG has been reported in individuals who faced traumatic circumstances

such as: natural disasters [41], terrorism [42], cancer [43], childhood sexual abuse [44], burn injuries [45], war [46], road trauma [47], and HIV/AIDS [48].

Calhoun, Tedeschi, Cann and Hanks [49] presented a Model of Growth in Grief to understand how losing "a close other" may result in the acknowledgment of a number of positive changes. Models of PTG focus on the degree to which the traumatic experience challenges world assumptions [50,51]. Deaths that are unexpected, less 'natural', and conflict with or disrupt an individual's assumptions and beliefs about the way the world should work (e.g., the death of a child) often result in more distress and PTG [51], as significant cognitive work is needed to reassess beliefs [49]. Rumination has an important role to play, as individuals attempt to reconstruct their beliefs following bereavement. Whilst this process can lay a foundation for growth and the process of rebuilding (deliberate rumination), it can also initially add to distress (intrusive rumination), and negatively impact PTG [52]. Disclosure plays an important role, with supported self-disclosure helping individuals to manage distress and rebuild challenged assumptions [49]. The Model of Growth in Grief [49] has been critiqued and a two-component model also proposed, where alongside the constructive aspects of PTG there is also an illusory facet to the response that means that PTG does not have a straightforward relationship to adjustment post-loss [53,54]. However, there continues to be wide acknowledgement that PTG can occur as a response to a wide range of traumatic experiences.

## Perinatal loss and posttraumatic growth

The possibility of PTG in the aftermath of pregnancy loss has received limited attention, with most studies grouping different types of perinatal losses together, resulting in calls for more research in this area [55,56]. Post-Traumatic Growth has been reported by parents following the death of their premature baby [57], with the change appearing greater in mothers (78%) than fathers (44%). Equally, an ethnographic study by Black and Sandelowski [58] found that parents expressed a new appreciation for life following a severe fetal anomaly diagnosis. Wright [59] used grounded theory with a sample of nineteen women who had experienced loss in pregnancy (between 8 and 40 weeks gestation) and found women felt more loving, compassionate, and appreciative of the relationships they had. More recently, Krosch and Shakespeare-Finch [60] reported moderate levels of PTG in women who had experienced miscarriage or stillbirth. However, so far, PTG has not been investigated in detail following pregnancy loss and no study has explicitly compared PTG in SB and EM, although Freedle and Kashubeck-West [61] found that gestational age was correlated with the Posttraumatic Growth Inventory [PTGI; 62] in their study in women following pregnancy loss.

## Current study

The Model of Growth in Grief is of particular interest when considering loss in pregnancy, as the majority of published findings of positive changes following loss have focused on the loss of a loved one with whom one had a longstanding relationship. Although mothers have been reported to form attachments early on in their pregnancy [63], mothers who experience loss in pregnancy do not have the opportunity to create direct life experiences with their baby, unlike in the death of other close family members [32]. Moreover, particular factors relating to the Model of Growth in Grief (challenge to assumptive beliefs, rumination and disclosure) may differ in the experience of EM and SB and result in differing levels of PTG.

To our knowledge, no study has so far tested the Model of Growth in Grief using between-group comparisons in the perinatal period. This study investigated factors associated with PTG in mothers following SB compared to mothers following EM, overcoming the limitations of previous studies and building on the recommendations of Freedle and Kashubeck-West

[61]. It was hypothesised that mothers would experience more challenge to assumptive beliefs following SB compared to EM, as it is widely known that rates of pregnancy loss are higher in the first 12 weeks. Similarly, it was hypothesised that levels of actual disclosure (discussing the loss) would be higher following SB compared to EM, because more of the mother's social network would be aware of the pregnancy. As the Model of Growth in Grief suggests that challenge to assumptive beliefs and disclosure both contribute to the development of PTG, it was also hypothesised that levels of PTG would be higher in mothers experiencing a SB compared to an EM. The study also aimed to investigate how theoretically-derived variables of the Model of Growth in Grief (challenge to assumptive beliefs, rumination and disclosure) explain unique variance in PTG when key factors are controlled for.

## Method

### Design

A cross-sectional between-groups design was adopted to compare PTG in mothers following SB and EM. The independent variable was type of loss (SB/EM) and primary dependent variable was PTG, with secondary dependent variables identified as key components from the Model of Growth in Grief (challenge to assumptive beliefs, rumination and disclosure). Ethical approval was granted by the University of Bath Psychology Ethics Committee (17–044).

### Procedure

Women were recruited online through advertisements placed on social media sites/accounts of perinatal death charities. Advertisements included a link directing participants to the study information sheet and online survey. Questionnaires were hosted on the Bristol Online Survey platform, enabling mothers to participate anonymously online. Participants viewed an information sheet that included information about the possible disadvantages or risks of taking part. Participants completed a consent form to participate in the study prior to completing questionnaires. They were automatically redirected to an exit page if they did not endorse the consent items. For all questions referring to 'the event' or 'the incident', participants were asked to think about 'losing my baby'. If participants had experienced multiple losses within the previous two to six years, they were asked to answer the questions in relation to the loss (SB/EM) that affected them the most. All data provided was anonymous. Participants were advised to contact their General Practitioner (GP) if they experienced distress following completion of the study or wished to discuss issues further. Participants were also given the option to contact the research team to arrange a phone call if they had concerns about their participation. At the end of the study questionnaires, participants consented to submit their responses and agreed that once they had submitted their survey they would be unable to request for their data to be withdrawn (due to the survey being anonymous). Completing the survey took 30 minutes.

### Participants

Women were eligible to participate if they spoke English, were aged over 18 years old and had experienced either a SB at 24 weeks gestation or later [UK definition; 5] or an EM before 13 weeks gestation within the last two to six years. This time point was used previously by Büchi, Mörgeli, Schnyder, Jenewein, Hepp, Jina, et al. [57] and was established as sufficient time for PTG to have occurred in their study of PTG in couples following the loss of their premature baby. Women who had experienced a late miscarriage (between the 13th and 23rd week of pregnancy) were not included in this study due to concerns regarding feasibility, time constraints, and greater heterogeneity across late miscarriage than in the EM or SB populations.

The sample comprised 57 mothers in the SB group and 63 mothers in the EM group. The mean age of participants was 34.74 years ($SD$ = 4.71) following SB and 36.71 years ($SD$ = 5.36) following EM. The majority of participants self-identified their ethnic background as white British, were married/in a civil partnership or cohabiting and educated to degree qualification or above. Living children were reported in 96.5% of the SB group ($N$ = 55) and 87.3% of the EM group ($N$ = 55). Eleven participants in the SB group (19.3%) reported being pregnant at the time of completing the study compared to six participants in the EM group (9.5%). The mean time since loss was 3.23 years ($SD$ = 1.13) following SB and 3.19 years ($SD$ = 1.09) following EM. A total of 42.1% of participants in the SB group ($N$ = 24) and 57.1% in the EM group ($N$ = 36) had previously experienced loss in pregnancy. For further demographic information and obstetric history of the sample, see Table 1.

## Materials

The Posttraumatic Growth Inventory [PTGI; 62] with 21-items measures positive existential growth following traumatic events. Items are rated on a Likert scale from 0 (*no change)* to 5 (*a very great degree of change)*. The total score ranges from 0–105, where higher scores indicate more growth. The phrase 'your crisis' in the instruction was changed to 'losing your baby'. The PTGI has good internal consistency and moderate test-retest reliability [62]. Cronbach's α for this study was .94.

The Posttraumatic Stress Disorder Checklist for DSM-5 [PCL-5; 64] has 20 items corresponding to DSM-5 criteria. For each item, a score of two or above (range 0 to 4) is regarded as clinically relevant. Participants were asked to rate each problem with respect to 'losing my baby'. Cronbach's α for this study was .96.

The Core Beliefs Inventory [CBI; 65] has nine items and participants rate the degree to which a recent highly stressful event led them to re-examine a number of core assumptions about themselves and their world. The total score range is 0–45 and items are rated on a Likert scale from 0 (*not at all)* to 5 (*a very great degree)*. The Cronbach's α for this study was .92.

The Event Related Rumination Inventory [ERRI; 66] has two subscales containing ten statements relevant to intrusive rumination (IR) and ten relevant to deliberate rumination (DR). The total score range is 0–60 and items are rated on a Likert scale from 0 (*not at all)* to 3 (*often)*. Instructions were changed from 'After an experience like the one you reported' to 'After a loss in pregnancy'. Cronbach's α reliability coefficients for the subscales were .95 (IR), .92 (DR) and .95 for the total.

The Disclosure of Trauma Questionnaire [DTQ; 67] has 34-item items rated on a Likert scale from 0 (*not at all)* to 3 (*completely)*, with three subscales: 'reluctance to talk', 'urge to talk', and 'emotional reactions'. The instructions were expanded to include 'in relation to losing your baby'. Cronbach's α for the subscales were .81 (reluctance to talk), .88 (urge to talk), .87 (emotional reactions), and .89 for the total.

In the absence of suitable formal published measures of actual self-disclosure, two questions were developed by the authors. Participants were asked to estimate the number of hours at three time points (first month, first year, and second year after loss) that they had spent talking to others about their feelings about losing their baby. They were also asked whether they had talked enough about their feelings at these time points. Actual self-disclosure was calculated by combining the total hours participants spent talking about their feelings about losing their baby in the first and second year after their loss.

The Perinatal Grief Scale Short Version [PGS; 68] has 33 items using Likert-type responses from 1 (*strongly agree)* to 5 (*strongly disagree)*. Higher scores reflect more grief, with a total score range of 33–165. Cronbach's α for this study was .94.

**Table 1. Sample characteristics (N = 120).**

| | Stillbirth | | Early Miscarriage | | *p* |
|---|---|---|---|---|---|
| | **Mean or N** | **SD or %** | **Mean or N** | **SD or %** | |
| Age (years) | 34.74 | 4.71 | 36.71 | 5.36 | .035 |
| Ethnicity | | | | | .392 |
| White British | 53 | 93 | 57 | 90.5 | |
| Other White background | 4 | 7 | 3 | 4.8 | |
| Multiple/Mixed ethnic group | - | - | 2 | 3.2 | |
| Other | - | - | 1 | 1.6 | |
| Marital Status | | | | | .410 |
| Single, never married | - | - | 1 | 1.6 | |
| Married/civil partnership or cohabiting | 56 | 98.2 | 59 | 93.7 | |
| Divorced or separated | 1 | 1.8 | 3 | 4.8 | |
| Employment Status | | | | | .511 |
| Full-time | 14 | 24.6 | 18 | 28.6 | |
| Part-time | 26 | 45.6 | 24 | 38.1 | |
| Student | - | - | 2 | 3.2 | |
| Homemaker | 16 | 28.1 | 16 | 25.4 | |
| Other | 1 | 1.8 | 3 | 4.8 | |
| Education | | | | | .485 |
| Left school without qualifications | - | - | 1 | 1.6 | |
| GCSE qualifications or equivalent | 4 | 7 | 5 | 7.9 | |
| A-Level or equivalent | 11 | 19.3 | 8 | 12.7 | |
| Degree qualification or above | 38 | 66.7 | 48 | 76.2 | |
| Prefer not to say | 1 | 1.8 | - | - | |
| Other | 3 | 5.3 | 1 | 1.6 | |
| Gestation of pregnancy loss | | | | | - |
| 1–4 weeks | - | - | 2 | 3.2 | |
| 5–8 weeks | - | - | 33 | 52.4 | |
| 9–12 weeks | - | - | 28 | 44.4 | |
| 24–27 weeks | 10 | 17.5 | - | - | |
| 28–31 weeks | 7 | 12.3 | - | - | |
| 32–35 weeks | 6 | 10.5 | - | - | |
| 36–40 weeks | 23 | 40.4 | - | - | |
| Over 40 weeks | 11 | 19.3 | - | - | |
| Previous Pregnancy Loss | | | | | .100 |
| Early Miscarriage | 23 | 40.4 | 34 | 54 | .136 |
| Late Miscarriage | 1 | 1.8 | 9 | 14.3 | .013 |
| Stillbirth | 2 | 3.5 | 2 | 3.2 | .919 |
| Currently pregnant | 11 | 19.3 | 6 | 9.5 | .134 |

*Note. SD* standard deviation; *N* number of participants; *p* p-value.

## Data analysis

Survey data was analysed using IBM SPSS Statistics version 23. Missing data points were identified in the actual disclosure variable, where 15 participants had not been able to estimate the amount of time they had spent talking about losing their baby. As this variable was considered very susceptible to individual differences, the missing data points were not substituted. All variables were assessed for outliers and violations of normality. Where assumptions of normality

were violated, equal variances were not assumed when testing for significance. Pearson's chi-square analyses and independent-samples t-tests were conducted to examine between-group differences that might confound subsequent analyses. A hierarchical stepwise regression analysis was conducted to investigate how theoretically-derived variables of the Model of Growth in Grief (challenge to assumptive beliefs, rumination, disclosure) explained unique variance in PTG when key factors were controlled. The variables were entered in four steps in line with the development of PTG through the Model of Growth in Grief: 1) Confounds—age, type of loss (SB vs EM), time since loss, perinatal grief and PTSD symptoms; 2) Challenge to assumptive beliefs; 3) Rumination; 4) Disclosure.

## Results

There were no significant group differences regarding ethnicity, marital status, education, employment, being currently pregnant or having had a previous loss in pregnancy (Table 1). However, more participants in the EM group (14.3%; $N = 9$) compared to SB (1.8%; $N = 1$) had previously experienced a late miscarriage ($p = .013$). Women in the SB group were significantly younger than those in the EM group (p = .035).

Mean scores and standard deviations are provided in Table 2. The SB group had significantly more perinatal grief (p < .001, d = 0.62) and PTSD symptoms (p = .034, d = 0.39) compared to EM. Compared to EM, the SB group had significantly greater PTG (p = .002, d = 0.58). Significant differences were found in intrusive rumination (p < .001, d = 0.68) and deliberate rumination (p = .043, d = 0.38) for SB compared to EM. There was also a significant difference in challenge to assumptive beliefs for SB compared to EM (p < .001, d = 0.94). Significant group differences in urge to talk (p = .007, d = 0.50) and emotional reactions during disclosure (p = .015, d = 0.45) were found. There was no significant difference in reluctance to talk between the two groups (p = ns). Compared to EM, the SB group had significantly more actual self-disclosure (p < .001, d = 1.08). In the first month and year after losing their baby, a

**Table 2. Means and standard deviations for questionnaire scores by type of loss.**

|  | Stillbirth | | Early Miscarriage | | t |
|---|---|---|---|---|---|
|  | **Mean** | **SD** | **Mean** | **SD** |  |
| PGS (perinatal grief) | 98.05 | 24.30 | 81.97 | 27.72 | 3.36*** |
| PCL5 (PTSD symptoms) | 29.42 | 19.30 | 21.51 | 21.03 | 2.15* |
| DTQ (disclosure) | 42.11 | 16.97 | 33.44 | 15.72 | 2.90** |
| DTQ–reluctance to talk | 13.16 | 8.66 | 11.44 | 8.89 | 1.07 |
| DTQ–urge to talk | 13.68 | 7.36 | 10.00 | 7.29 | 2.75** |
| DTQ–emotional reactions | 15.26 | 7.70 | 12.00 | 6.82 | 2.46* |
| Actual Self-disclosure (hours) | 287.79 | 317.78 | 38.95 | 69.27 | 5.32*** |
| ERRI (rumination) | 35.84 | 11.90 | 27.52 | 16.05 | 3.20** |
| ERRI–intrusive rumination | 17.88 | 6.77 | 12.62 | 8.58 | 3.70*** |
| ERRI–deliberate rumination | 17.96 | 7.34 | 14.90 | 8.89 | 2.04* |
| CBI (challenge to assumptive beliefs) | 24.04 | 9.08 | 14.79 | 10.49 | 5.17*** |
| PTGI (posttraumatic growth) | 45.96 | 23.31 | 32.11 | 24.20 | 3.19** |

*Note. SD* standard deviation, *t* t-value, *PGS* Perinatal Grief Scale Short Version, *PCL-5* Posttraumatic Stress Disorder Checklist for DSM-5, *DTQ* Disclosure of Trauma Questionnaire, *ERRI* Event Related Rumination Inventory, *CBI* Core Beliefs Inventory, *PTGI* Posttraumatic Growth Inventory.

* $p < .05$,

** $p < .01$,

*** $p < .001$.

**Table 3. Bivariate correlations for both stillbirth and early miscarriage groups.**

| Measure | PTGI | Age | Type of Loss | Time since Loss | PCL-5 | PGS | CBI | ERRI Int. | ERRI Delib | DTQ Urge to Talk | DTQ Reluc to Talk | Actual Self-Discl |
|---|---|---|---|---|---|---|---|---|---|---|---|---|
| PTGI | - | -0.59 | -.28*** | -.11 | .08 | .13 | .47*** | .16* | .30*** | .54*** | -.19* | .37*** |
| Age | | - | .19* | .08 | -.10 | -.10 | -.05 | -.07 | .05 | -.04 | -.13 | -.09 |
| Type of Loss | | | - | -.02 | -.19* | -.30** | -.43*** | -.32*** | -.19* | -.25** | -.10 | -.49*** |
| Time since Loss | | | | - | -.07 | -.03 | -.01 | -.12 | -.08 | -.12 | .17 | -.05 |
| PCL-5 | | | | | - | .79*** | .46*** | .76*** | .56*** | .27** | .47*** | .06 |
| PGS | | | | | | - | .56*** | .68*** | .50*** | .30** | .43** | .14 |
| CBI | | | | | | | - | .37*** | .39*** | .39*** | .33*** | .33*** |
| ERRI Int | | | | | | | | - | .61*** | .34*** | .35*** | .13 |
| ERRI Del | | | | | | | | | - | .45*** | .13 | .12 |
| DTQ Urge | | | | | | | | | | - | -.19* | .20* |
| DTQ Reluc | | | | | | | | | | | - | -.12 |
| Actual Self-Discl | | | | | | | | | | | | - |

*Note. PTGI* Posttraumatic Growth Inventory, *PCL-5* Posttraumatic Stress Disorder Checklist for DSM-5, *PGS* Perinatal Grief Scale Short Version, *CBI* Core Beliefs Inventory, *ERRI* Event Related Rumination Inventory, *DTQ* Disclosure of Trauma Questionnaire.

* $p < .05$,

** $p < .01$,

*** $p < .001$.

greater percentage of participants felt they did not talk enough about their EM (60.3% in first month ($N = 38$); 55.6% in first year ($N = 35$)) compared to participants following SB (47.4% in first month ($N = 27$); 49.1% in first year ($N = 28$)). In contrast, more participants following SB (59.6%, $N = 34$) reported that they did not talk enough in the second year after losing their baby compared to participants following EM (46%, $N = 29$).

PTG was found to be significantly correlated with type of loss (p = .001), challenge to assumptive beliefs (p < .001), deliberate rumination (p < .001), intrusive rumination (p = .041), urge to talk (p < .001), reluctance to talk (p = .019), and actual self-disclosure (p < .001), as indicated in Table 3. The following variables were not significantly correlated with PTG: PTSD symptoms, perinatal grief, depression, anxiety, emotional reactions, previous loss in pregnancy, experiencing multiple losses within the last two to six years, having living children, being pregnant now, having a child since experiencing a pregnancy loss, age and time since pregnancy loss (all p = NS; see Table 3 for correlations of key variables).

Results of the hierarchical stepwise regression analysis are outlined in Table 4. Overall, 9.6% of variance in PTG was explained from step 1 (age, type of loss, time since loss, perinatal grief and PTSD symptoms). Challenge to assumptive beliefs predicted an additional 17.5% in step 2. Intrusive rumination and deliberate rumination predicted 5.5% of variance in PTG in step 3 and urge to talk, actual self-disclosure, and reluctance to talk predicted a further 15.3% in step 4. The final model explained 47.9% of variance in PTG, $F(11,91) = 7.61$, p < .001. Urge to talk ($\beta = .33$) and challenge to assumptive beliefs ($\beta = .43$) significantly predicted PTG but the other variables no longer made a unique contribution.

## Discussion

This study compared levels of PTG in mothers following SB or EM and examined whether theoretically-derived variables of the Model of Growth in Grief explained unique variance in PTG

**Table 4. Multiple regression analysis with posttraumatic growth as dependent variable.**

| | B | SE B | β | t | p |
|---|---|---|---|---|---|
| Step 1 | | | | | |
| Constant | 61.26 | 21.94 | | 2.79 | .01 |
| Age | 0.32 | .48 | .01 | .07 | .95 |
| Type of Loss | -13.28 | 5.07 | -.27 | -2.62 | .01 |
| Time since Loss | -2.69 | 2.17 | -.12 | -1.24 | .22 |
| Perinatal grief | .07 | .15 | .08 | .51 | .61 |
| PTSD symptoms | -.06 | .19 | -.05 | -.30 | .76 |
| Step 2 | | | | | |
| Constant | 49.46 | 19.95 | | 2.48 | .02 |
| Age | -.11 | .43 | -.02 | -.24 | .81 |
| Type of Loss | -5.32 | 4.86 | -.11 | -1.10 | .28 |
| Time since Loss | -2.66 | 1.96 | -.12 | -1.36 | .18 |
| Perinatal grief | -.12 | .14 | -.13 | -.88 | .38 |
| PTSD symptoms | -.12 | .17 | -.10 | -.69 | .49 |
| Challenge to assumptive beliefs | 1.22 | .25 | .54 | 4.81 | .00 |
| Step 3 | | | | | |
| Constant | 44.60 | 19.79 | | 2.25 | .03 |
| Age | -.28 | .42 | -.06 | -.65 | .51 |
| Type of Loss | -3.93 | 4.92 | -.08 | -.80 | .43 |
| Time since Loss | -2.21 | 1.92 | -.10 | -1.15 | .25 |
| Perinatal grief | -.16 | .14 | -.17 | -1.14 | .26 |
| PTSD symptoms | -.32 | .20 | -.26 | -1.60 | .11 |
| Challenge to assumptive beliefs | 1.17 | .25 | .51 | 4.60 | .00 |
| Deliberate Rumination | .78 | .33 | .26 | 2.37 | .02 |
| Intrusive Rumination | .25 | .45 | .08 | .55 | .58 |
| Step 4 | | | | | |
| Constant | 28.51 | 18.54 | | 1.54 | .13 |
| Age | -.29 | .38 | -.06 | -.75 | .46 |
| Type of Loss | 1.40 | 4.79 | .03 | .29 | .77 |
| Time since Loss | -.63 | 1.77 | -.03 | -.35 | .72 |
| Perinatal grief | -.14 | .12 | -.15 | -1.11 | .27 |
| PTSD symptoms | -.17 | .18 | -.15 | -.97 | .34 |
| Challenge to assumptive beliefs | .98 | .25 | .43 | 3.95 | .00 |
| Deliberate Rumination | .32 | .31 | .11 | 1.04 | .30 |
| Intrusive Rumination | .20 | .40 | .07 | .51 | .61 |
| Urge to Talk | 1.07 | .32 | .33 | 3.34 | .00 |
| Actual self-disclosure | .02 | .01 | .17 | 1.82 | .07 |
| Reluctance to Talk | -.44 | .29 | -.16 | -1.50 | .14 |

*Note.* B unstandardised beta, SE B standard error for unstandardised beta, β standardised beta (regression coefficient), t t-value, p p-value.

Note $R^2$ = .096 for step 1; $\Delta R^2$ = .175 for step 2 ($p < .001$); $\Delta R^2$ = .055 for step 3 ($p < .05$); $\Delta R^2$ = .153 for step 4 ($p < .001$).

when key factors were controlled for. Results indicated that mothers who had experienced a SB experienced significantly greater PTG, PTSD symptoms, and perinatal grief than mothers who had experienced an EM. Greater challenge to assumptive beliefs, intrusive and deliberate rumination, urge to talk, and reported actual self-disclosure were also found in

mothers following SB compared to EM. The final regression model explained 47.9% of variance in PTG.

Our results showing higher levels of PTSD symptoms and PTG following SB compared to EM are in line with previous research that found that pregnancy length was linked to PTSD severity [16], although a review of reproductive loss highlighted that other factors (e.g., younger age, lower education, and a history of other traumas/mental health problems) also contribute to the risk of developing PTSD [69]. The differences in PTSD symptoms between SB and EM may have been due to the greater physical trauma and perceived risk in SB, as well as the differences in prenatal attachment and the loss becoming more tangible as time progressed in pregnancy [70]. Previous studies have reported PTG in mothers following SB [71], miscarriage [72], and pregnancy loss in general [60]. However, this is the first study we are aware of that explicitly compared PTG in a SB and EM population. Krosch and Shakespeare-Finch [60] found higher levels of PTG following pregnancy loss (M = 51.22, SD = 20.13) than in the present study, which may be due to their longer time since loss (M = 4.01 years compared to 3.23 years (SB) and 3.19 years (EM) in this study). There continues to be a paucity of research on the timeframe required for PTG to develop.

Participants reported moderate to high levels of perinatal grief following their loss. Scores were consistent with previous research [73] and findings that bereaved parents can experience grief for many years following their loss [74,75]. Emerging models of perinatal bereavement suggest intense distress and grief are experienced in the short term and the most intense reactions typically decrease within the first 12 months (and significantly two years) after loss [76]. Consistent with previous research [77], higher grief scores were found in the SB group, highlighting the distressing nature of this type of loss.

Significant group differences regarding challenge to assumptive beliefs were found, suggesting SB is associated with a greater disruption of mothers' beliefs about the way the world should work than in EM. These findings are consistent with theory [49,51] and previous research highlighting the differences in challenge to assumptive beliefs by gestational age [60].

Over half of mothers following EM felt they did not talk enough about their loss in the first month and year following their miscarriage, mirroring previous reports that EM remains shrouded in shame and silence, and the difficulty mothers have in talking about their loss [78]. Mothers following SB disclosed greater urge to talk and reported greater actual self-disclosure than mothers following EM. Crawley, Lomax and Ayers [79] emphasised the importance of sharing memories of the stillborn baby to aid adjustment and wellbeing, and disclosure appears to have been an important factor in the development of PTG in this study. However, the differences in disclosure may also reflect a lesser need for some mothers to talk about their EM than their SB, potentially reflecting a reduced need to reconstruct beliefs following loss at different stages of pregnancy. Future research would benefit from developing a validated measure of disclosure to further test the Model of Growth in Grief.

Individual differences, rather than the nature of the trauma (SB vs EM), were expected to determine rumination. However, significant differences in intrusive rumination and deliberate rumination were found between the two groups and may be the result of the differences in the visibility of the pregnancy, time spent pregnant, and expectations women hold of experiencing SB compared to EM. On average, mothers who had experienced an EM revealed higher levels of deliberate rumination than intrusive rumination. This is consistent with reports that intrusive thoughts are likely to occur in the immediate aftermath of an event, with deliberate thoughts occurring after the initial shock and distress subside [55]. It is therefore interesting that levels of intrusive rumination and deliberate rumination were comparable in mothers two to six years after experiencing a SB, which is a novel finding, and may relate to the physical trauma of SB.

A major strength of this study is the use of a between-groups design to investigate PTG in two groups of mothers following pregnancy loss (SB/EM), overcoming the limitations of previous correlational studies. Another strength is the application of theoretically-derived variables of the Model of Growth in Grief (including disclosure), to our knowledge the first of its kind in relation to pregnancy loss. The findings provide support for the relevance of the variables in the Model of Growth in Grief [49] in the development of PTG in mothers following SB and EM. Finally, this study had a well-powered sample and used validated measures to capture key variables identified in the model.

This study has some limitations. In the absence of high-quality validated measures of actual disclosure (a problem encountered in other studies of PTG [80], a single-item measure of actual self-disclosure was created for this study, which may not have been sufficient to accurately measure the construct; however, in the design of this measure, attempts were made to help structure recall of time spent talking about loss. The large variability in responses may demonstrate problems with reliability and validity, but may also be an accurate reflection of individual differences in self-disclosure. Additionally, although the PTGI is the most established measure of PTG, there is some research to suggest that this may measure a subjective perception of growth rather than actual growth [81], although findings are mixed and other research suggests that self-reported growth on the PTGI is corroborated [82]. As the sample is relatively homogenous in terms of ethnicity, marital and educational status, generalisation of the findings to other populations may be limited. In addition, recruiting from a perinatal loss charity means that the research was more likely to recruit participants who were struggling with their loss or interested in reflecting on their experience. Furthermore, participants were not asked how many losses they had experienced in their lifetime or how long ago these losses had occurred.

The proportion of variance (47.9%) in PTG accounted for in this study indicates there are other factors contributing to its occurrence in mothers following SB and EM not studied here. The model suggests that the development of PTG will take time and although this study replicated the two to six year time point used previously [57], there is a lack of understanding in the literature of how time affects PTG and when levels may peak. Finally, a greater percentage of mothers following EM had also experienced a LM (14.3%) compared to mothers following SB (1.8%), and the EM group were slightly older than the SB group; these differences may have confounded results.

Interventions targeting the key variables in the Model of Growth in Grief (challenge to assumptive beliefs, disclosure and rumination) are likely to be clinically useful to promote psychological adjustment in mothers who have experienced SB and EM and are relevant to practitioners working in this area. Whilst PTG is common following trauma, it is not universal, and practitioners should not expect that every mother will experience growth or that it is necessary for psychological adjustment, and some bereaved mothers may find the concept of growth offensive [83]. Where mothers' beliefs have been seriously challenged, clinicians could work to support mothers to talk about their feelings and encourage greater disclosure with friends and family. Supporting mothers through the process of reassessing such core beliefs may help to lay a foundation for growth and the process of rebuilding [65]. Ultimately, there remains a need to change attitudes to pregnancy loss and disclosure at a societal level, rather than simply supporting mothers and their families.

## Conclusion

Significantly higher levels of PTG, PTSD symptoms and perinatal grief were found in mothers following SB compared to EM. Mothers experienced a greater challenge to their assumptive

beliefs and revealed higher levels of disclosure following SB. These findings can partially be explained by differences in key variables from the Model of Growth in Grief. It is likely that addressing these factors will help to alleviate psychological distress and promote the development of PTG.

## Acknowledgments

The authors thank the stillbirth and neonatal death charity (Sands, UK) for their advice and support throughout this project. We would also like to thank all the participating mothers for contributing to this study.

## Author Contributions

**Conceptualization:** Kirsty Ryninks, Megan Wilkinson-Tough, Sarah Stacey, Antje Horsch.

**Data curation:** Kirsty Ryninks.

**Formal analysis:** Kirsty Ryninks, Megan Wilkinson-Tough, Antje Horsch.

**Investigation:** Kirsty Ryninks, Megan Wilkinson-Tough, Antje Horsch.

**Methodology:** Kirsty Ryninks, Megan Wilkinson-Tough, Sarah Stacey, Antje Horsch.

**Project administration:** Kirsty Ryninks.

**Resources:** Sarah Stacey.

**Supervision:** Megan Wilkinson-Tough, Sarah Stacey, Antje Horsch.

**Writing – original draft:** Kirsty Ryninks.

**Writing – review & editing:** Kirsty Ryninks, Megan Wilkinson-Tough, Sarah Stacey, Antje Horsch.

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
