## [Decision Letter · Decision Letter 0]

27 Apr 2022

PONE-D-22-01427Comparing posttraumatic growth in mothers after stillbirth or early miscarriagePLOS ONE

Dear Dr. Stacey,

Thank you for submitting your manuscript to PLOS ONE. After careful consideration, we feel that it has merit but does not fully meet PLOS ONE’s publication criteria as it currently stands. Therefore, we invite you to submit a revised version of the manuscript that addresses the points raised during the review process.

We look forward to receiving your revised manuscript.

Kind regards,

David Desseauve, MD, MPH, PhD

Academic Editor

PLOS ONE

Journal Requirements:

Reviewers' comments:

Reviewer's Responses to Questions

**Comments to the Author**

1. Is the manuscript technically sound, and do the data support the conclusions?

Reviewer #1: Yes

Reviewer #2: Yes

2. Has the statistical analysis been performed appropriately and rigorously? 

Reviewer #1: I Don't Know

Reviewer #2: Yes

3. Have the authors made all data underlying the findings in their manuscript fully available?

Reviewer #1: Yes

Reviewer #2: Yes

4. Is the manuscript presented in an intelligible fashion and written in standard English?

Reviewer #1: Yes

Reviewer #2: Yes

5. Review Comments to the Author

Reviewer #1: This well written article compares posttraumatic growth in women after stillbirth or early miscarriage in 120 women. The manuscript is logic and easy to follow and comes to sound conclusions. It addresses a topic where one still needs more knowledge.

A few uncertainties need attention:

1. Table 1. Line 261 refers to ‘being currently pregnant’ but it is not reflected in the table.

2. Table 2. The legend refers to ‘GAD-7’ and ‘PHQ’ but it is not in the table.

3. The words in brackets following ’Model of Growth in Grief’ differ. Please ompare line 157 with line 141. The meaning of ‘CAB’ is unclear.

Reviewer #2: Thanks for this submission.

In my hospital, all women with SB could benefit from a psychiatrist consultation. I guess it is a way to incresase the rate of PTG. In the other hand, women with early miscarriage don't benefit from that.

Did you explore psychiatrist intervention in your population?

6. PLOS authors have the option to publish the peer review history of their article (what does this mean?). If published, this will include your full peer review and any attached files.

Reviewer #1: No

Reviewer #2: No

---

## [Author Response · Author response to Decision Letter 0]

17 May 2022

Dear Editor and reviewers,

Many thanks for your review of our manuscript, and thoughtful comments and feedback. Please see copied below your comments and our responses.

Kind regards,

Sarah Stacey (on behalf of all co-authors)

Reviewers' comments:

Reviewer #1: 

This well written article compares posttraumatic growth in women after stillbirth or early miscarriage in 120 women. The manuscript is logic and easy to follow and comes to sound conclusions. It addresses a topic where one still needs more knowledge.

A few uncertainties need attention:

Many thanks for highlighting these, our comments are below.

1. Table 1. Line 261 refers to ‘being currently pregnant’ but it is not reflected in the table.

Response: Many thanks for highlighting this. We have now included the data for the number of participants in each group who were pregnant at the time of the study into Table 1.

2. Table 2. The legend refers to ‘GAD-7’ and ‘PHQ’ but it is not in the table.

Response: Many thanks for spotting this. We have deleted the reference to these questionnaires from the legend.

3. The words in brackets following ’Model of Growth in Grief’ differ. Please compare line 157 with line 141. The meaning of ‘CAB’ is unclear.

Response: With thanks again. We have removed the acronym and amended line 157 to be the same as line 141. Both lines now reflect that the current study particularly explored experiences of challenge to assumptive beliefs (CAB), rumination and disclosure in relation to early miscarriage and stillbirth.

Reviewer #2: 

Thanks for this submission.

In my hospital, all women with SB could benefit from a psychiatrist consultation. I guess it is a way to incresase the rate of PTG. In the other hand, women with early miscarriage don't benefit from that.

Did you explore psychiatrist intervention in your population?

Many thanks for your insightful question. We did not explore the possibility of impact of psychiatrist intervention in the current study, although agree that this could be worthy of further investigation. This study was conducted online so participants will have been treated at many different hospitals and, therefore, will likely have received differing kinds of care after their loss. 

Psychiatrist intervention could be considered for any hospital patient who was thought to be experiencing severe mental health difficulties and could be accessed through the hospital-based Liaison Psychiatry service which exists in most larger acute hospitals in England. We did not gather data as to whether any of the participants in this study were referred to this kind of service at the time of their loss.

---

## [Editor Report · Decision Letter 1]

29 Jun 2022

Comparing posttraumatic growth in mothers after stillbirth or early miscarriage

PONE-D-22-01427R1

Dear Dr. Stacey,

We’re pleased to inform you that your manuscript has been judged scientifically suitable for publication and will be formally accepted for publication once it meets all outstanding technical requirements.

Kind regards,

David Desseauve, MD, MPH, PhD

Academic Editor

PLOS ONE

---

## [Editor Report · Acceptance letter]

29 Jul 2022

PONE-D-22-01427R1 

Comparing posttraumatic growth in mothers after stillbirth or early miscarriage 

Dear Dr. Stacey:

I'm pleased to inform you that your manuscript has been deemed suitable for publication in PLOS ONE. Congratulations! Your manuscript is now with our production department. 

Kind regards, 

on behalf of

Dr. David Desseauve 

Academic Editor

PLOS ONE